# From Stable to Lab—Investigating Key Factors for Sudden Deaths Caused by *Streptococcus suis*

**DOI:** 10.3390/pathogens8040249

**Published:** 2019-11-20

**Authors:** Isabel Hennig-Pauka, Rabea Imker, Leonie Mayer, Michael Brügmann, Christiane Werckenthin, Heike Weber, Andrea Menrath, Nicole de Buhr

**Affiliations:** 1Field Station for Epidemiology, University of Veterinary Medicine Hannover, 30559 Hannover, Germany; isabel.hennig-pauka@tiho-hannover.de (I.H.-P.); andrea.menrath@tiho-hannover.de (A.M.); 2Department of Physiological Chemistry, University of Veterinary Medicine Hannover, 30559 Hannover, Germany; rabea.imker@tiho-hannover.de; 3Research Center for Emerging Infections and Zoonoses (RIZ), University of Veterinary Medicine Hannover, 30559 Hannover, Germany; 4Institute of Bacteriology and Mycology, Centre for Infectious Diseases, Veterinary Faculty, University of Leipzig, 04109 Leipzig, Germany; leonie.mayer@vetmed.uni-leipzig.de; 5Food and Veterinary Institute Oldenburg, Lower Saxony State Office for Consumer Protection and Food Safety, 26029 Oldenburg, Germany; michael.bruegmann@laves.niedersachsen.de (M.B.); christiane.werckenthin@laves.niedersachsen.de (C.W.); 6Tierpark Nordhorn, 48531 Nordhorn, Germany; heike.weber@tierpark-nordhorn.de

**Keywords:** *Streptococcus suis*, sudden death, DIC, NETs, immune reaction

## Abstract

Swine stocks are endemically infected with the major porcine pathogen *Streptococcus* (*S.*) *suis*. The factors governing the transition from colonizing *S. suis* residing in the tonsils and the exacerbation of disease have not yet been elucidated. We analyzed the sudden death of fattening pigs kept under extensive husbandry conditions in a zoo. The animals died suddenly of septic shock and showed disseminated intravascular coagulopathy. Genotypic and phenotypic characterizations of the isolated *S. suis* strains, a tonsillar isolate and an invasive *cps* type 2 strain, were conducted. Isolated *S. suis* from dead pigs belonged to *cps* type 2 strain ST28, whereas one tonsillar *S. suis* isolate harvested from a healthy animal belonged to ST1173. Neither *S. suis* growth, induction of neutrophil extracellular traps, nor survival in blood could explain the sudden deaths. Reconstituted blood assays with serum samples from pigs of different age groups from the zoo stock suggested varying protection of individuals against pathogenic *cps* type 2 strains especially in younger pigs. These findings highlight the benefit of further characterization of the causative strains in each case by sequence typing before autologous vaccine candidate selection.

## 1. Introduction

*Streptococcus* (*S.*) *suis* is one of the main bacterial pathogens that causes high losses in nursery pigs and, furthermore, is an emerging zoonotic pathogen [1,2]. Various serotypes have been described, but in recent years, some serotypes have been reclassified as other streptococcal species [3,4,5,6]. Assessment of the virulence of strains is complex due to the high diversity within the various serotypes, and the fact that multiple serotypes can be isolated from the same animal [2,7,8,9,10]. Serotype 2 is most frequently isolated during disease outbreaks in pigs and humans [7], followed by serotypes 1, 9, and 7, which have been considered to represent more invasive isolates [11,12,13]. Alongside serotypes, the multilocus sequence typing scheme (MLST) has been used to describe the *S. suis* population structure and the diversity of isolates [14,15].

Control of *S. suis* infections in pigs without antibiotics is difficult because of a lack of adequate protection by vaccination on most farms [16]. 

Clinical manifestation of infection involves different pathological findings including meningitis, septicemia, arthritis, and sudden death [2,17,18,19]. In addition, *S. suis* is considered to be a secondary pathogen causing respiratory disease following, for example, viral infections with influenza virus and porcine reproductive and respiratory syndrome virus (PRRSV) [20]. As a physiological mucosal colonizer, *S. suis* belongs to the tonsillar microbiome of pigs [21]. The tonsillar microbiome includes bacterial species belonging to the phyla Firmicutes and Proteobacteria, including the families Enterobacteriaceae and Pasteurellaceae as well as streptococci, enterococci, and staphylococci [21]. High *S. suis* isolation rates of 98% in healthy pigs have been described [22].

While sites of clinical manifestation like the joints and brain are naturally sterile, at the colonization sites of *S. suis*, the bacterium is embedded in an ecosystem of other microorganisms. *S. suis* is able to spread systemically (lymphogenously or hematogenously) from the tonsils to the sites of clinical manifestation, but the mechanisms are not well understood [23]. Recently, the enteric route of infection was investigated in experimental infection studies and cell culture experiments, but further studies are needed to assess an oro-gastrointestinal *S. suis* infection route in piglets [24,25,26]. 

As soon as an infection occurs, the host starts fighting against *S. suis.* Therefore, as the first line of defense, neutrophils infiltrate the infected tissue. If antibodies are present, an opsonization can lead to phagocytosis of *S. suis.* In the absence of specific antibodies or the production of factors by *S. suis* that degrade immunoglobulins such as Ide_Ssuis,_ the efficiency of phagocytosis is diminished [27].

The anti-phagocytic effect of the capsule of *S. suis* is an important virulence-associated factor [2,28], but non-typeable and non-encapsulated strains can also be pathogenic and can also become capsulated after some passages or under specific environmental conditions [29,30]. Apart from phagocytosis, a release of neutrophil extracellular traps (NETs) has been described as another mechanism for activated neutrophils to counteract invading pathogens [31]. These NETs consist of a DNA backbone and are decorated with antimicrobial compounds such as antimicrobial peptides and granular proteins [31,32,33]. They can be induced by chemokines, pathogens, and products of pathogens including toxins [34,35,36]. Depending on the pathogen, NETs can entrap and partially kill bacteria, but several evading strategies of bacteria have been reported. One evasion strategy described for *S. suis* is the production of the DNases SsnA and EndAsuis, which degrade NETs [37,38] and help *S. suis* evasion. On the other hand, the host itself produces DNases to degrade NETs, as they are detrimental to the host. NETs have been described to be involved in autoimmune diseases [39,40,41,42,43], thrombosis [44], and disseminated intravascular coagulopathy (DIC) in sepsis [45]. 

The aim of this study was to understand the sudden death of pigs at a zoo via pathological findings of DIC and the isolation of *S. suis* from parenchymatous organs. Starting with the description of the clinical cases occurring under extensive husbandry conditions, potential trigger factors for disease have been described besides those routinely found on conventional farms. The genotypes and phenotypes of the pathogenic isolates, as well as commensal tonsillar isolates from the stock, were characterized and compared to a well characterized *S. suis* meningitis isolate. Results of in vitro experiments comparing the survival of isolates in the blood of two pig stocks, as well as in humans, will pave the way for understanding infection dynamics in stock and the implementation of specific methods for disease prevention. Lab findings in in vitro experiments helped to explain observed clinical findings and support the importance of deeper typing of *S. suis* field isolates beyond routine diagnostics for herd health management of *S.-suis*-related diseases. 

## 2. Results

### 2.1. Diagnosis of Streptococcal Disease in a Zoo Pig Stock 

At a zoo in the north-western part of Germany, sudden death without obvious clinical signs occurred in several pigs. Therefore, the veterinarians attempted to identify the reasons for the death of these pigs. The case history showed that the pigs were part of a breeding program for an ancient regional pig breed called Buntes Bentheimer (Figure 1A), involving seven breeding sows and two boars. Pigs were housed at the zoo under extensive husbandry conditions in a straw-lined, confined building with intermittent access to a large, sandy outdoor area (Figure 1B). Litters were kept together with the adult animals until three to four months of age, when they were transported to a fattening farm equipped with resources and space for farrowing and nursery 14 km away (Figure 1C). 

At the time of the disease outbreak, approximately 35 animals were kept at the zoo and 40 in the outlying fattening unit. Sudden death without obvious clinical signs occurred in this confined unit one week after transport of pigs at 30–40 kg (aged >12 weeks). Pigs of different age groups were kept in groups of four to eight animals, in a total of eight pens and in the same air-space. This unit was always occupied with pigs of different age groups because of the continuous pig flow from the zoo and sale of heavy fatteners for slaughter. For this reason, cleaning and disinfection were not routinely performed in this unit. Pigs, which had died overnight, were brought to the Food and Veterinary Institute, Oldenburg for diagnostics. Gross pathological findings were a generalized cutaneous hyperemia, and histological findings revealed a disseminated intravascular coagulopathy in several parenchymatous organs (Figure 2). *S. suis* was isolated from the lung, kidney, and spleen in a pure culture in large amounts. Findings are summarized in Table 1.

Within the following months, more pigs died in the outlying fattening unit. Due to a higher level of alertness from the animal keeper, clinical signs were recorded in some pigs before death. Pigs were reluctant to move, lay in a lateral position, and showed cyanosis and fevers of >40.5 °C. Diseased pigs were treated immediately with 20 mg/kg body weight of a long-acting amoxicillin, with three doses at two-day intervals. Most treated pigs recovered from the disease. In a five-month period from October to February a total of 12 pigs aged 3–7 months died, of which five pigs were sent for further diagnostics. However, streptococcal disease could not be confirmed by cultivation after the first two cases due to antibiotic treatment (Table 1). 

During this five-month period, a herd visit was performed in a phase without diseased pigs or sudden death in order to examine husbandry and management conditions and to take blood and tonsillar brushing samples from healthy pigs.

In total, 31 healthy pigs were sampled at the ages of 7 (n = 9), 14 (n = 7), and 22 (n = 15) weeks old. Tonsillar brushing samples were examined bacteriologically for *S. suis* by routine diagnostic methods, resulting in only two positive samples in the group of 22-week-old pigs (Table 2). The isolated tonsillar strains are hereinafter referred to as T17 andT28. 

In parallel, oral fluid sampling was performed with ropes in the fattening unit to harvest material for immunization of sows and their younger piglets at the zoo. Within 36 h after exposure of animals at the zoo to the saliva-containing ropes, two pigs aged two months became diseased and had to be treated. Diseased animals recovered and no further clinical signs were observed. No further sudden death or disease due to *S. suis* occurred after animal exposure to saliva-containing ropes.

Nevertheless, as descriptions of DIC in pigs due to *S. suis* are rare, we performed further studies to gain a comprehensive understanding of the pathogenesis in this clinical case. 

### 2.2. Detection of Markers for Neutrophil Extracellular Traps (NETs) in Inner Organs Infected with S. suis and DIC Diagnosis

As DIC has been described to be associated with the formation of neutrophil extracellular traps (NETs) [45,46], we performed in situ immunofluorescence analysis in the histological samples. In all tested internal organs (lung, kidney, spleen), *S. suis* was detectable in good correlation with the microbiological findings. Furthermore, DNA–histone-1 complexes, a classical marker for NETs, were present (Figure 3). Especially in the kidney (Figure 3C,D), signals for DNA–histone-1 complexes were detected in similar paracellular areas as the DIC in HE stainings (Figure 2D). Therefore, we investigated the genotypic and phenotypic characteristics of this strain in order to better understand the sudden death due to *S.-suis*-induced DIC with detection of NETs.

### 2.3. Genotypic Characterization of Isolated S. suis Strains

Strains 483 and 484 isolated from the inner organs were identified by multiplex PCR as capsular type 2 or 1/2 (*cps* type *2*) and were positive for the virulence factor muramidase-released protein (*mrp*) (Appendix A, Figure A1). No variant of *mrp* was detected by PCR analysis (Appendix A, Figure A2). Furthermore, they tested negative for suilysin (SLY, *sly*) and extracellular factor (EF, *epf*) including PCR-based analysis for *epf* variants (Table 2; Appendix A, Figure A3). As the multiplex PCR conducted did not differentiate between serotypes 2 and 1/2, we performed Sanger sequencing analysis of the *cpsK* locus as described previously [47]. An alignment of the sequence fragments from Strains 483 and 484 and *cps2K* (GenBank: AF118389.1) showed that the sequences were identical, in particular at the position 483 of the *cpsK* gene (G nucleotide). Therefore, Strains 483 and 484 were identified as capsular type 2 (Appendix A
Figure A4). The tonsillar isolates T17 and T28 were negative for all tested capsular types (*cps* 1, 2, 4, 7, and 9) and the virulence-associated genes *mrp, sly, epf.* A multilocus sequence typing (MLST) of Strains 483 and 484 resulted in sequence type (ST) 28, whereas for T17, the new ST1173 was identified. Detailed information is available in the PubMLST database with id:2130 (https://pubmlst.org/bigsdb? page=info&db= pubmlst_ssuis_ isolates&id=2130). 

*S. suis* Strains 483, 484, and T17 were further analyzed during this study and compared to a *S. suis* (Strain 10) meningitis isolate from a pig, which has been well characterized (serotype 2, ST1, *mrp*^+^, *sly*^+^, *epf*^+^) and frequently used in infection and mutagenesis studies [28,32,37,38,48,49]. This strain was included in the subsequent phenotypic analysis as it has been described as an invasive strain, the capsular type being identical to the isolates from the dead pigs.

### 2.4. Phenotypic Characterization of S. suis Isolates 

After the genotypic characterization of the *S. suis* isolates from the lung and spleen, and the confirmation of *S. suis* infection in the same organs by immunofluorescence microscopy (Figure 3A (lung) and 3B (spleen)), we began phenotypic characterization of the isolates. As a first step, we compared the growth of the isolates under different conditions to determine whether this could explain the rapid death of the pigs infected with Strains 483 or 484. Interestingly, although isolates 483 and 484 were similar with respect to their virulence-associated factors, they grew in a different way in Todd Hewitt Broth (THB) media. The differences in growth were significantly different over time and strain-dependently (p_time_ < 0.0001, p_strain_ < 0.02). The tonsillar isolate T17 and the isolate from the lung (483) reached the exponential and stationary growth phases in parallel, and no significant difference was detected. The isolates from the spleen (484) and the brain (strain 10) grew more slowly. They reached the stationary growth phase 1–2 h later, and significant differences in growth were detectable at 3 and 4 h between all strains except T17 and Strain 483 (Figure 4A). Further studies are needed to understand and explain the different growth patterns of Strains 483 and 484. The growth of the three tonsil isolates was compared and all grew in THB, but reached different final optical densities (Appendix A, Figure A5). 

Uptake by aerosols via the respiratory tract is a common route of *S. suis* infection. Therefore, we additionally analyzed the growth of the strains in bronchoalveolar lavage fluid (BALF) of healthy pigs, mimicking growth conditions during a lung infection. All tested isolates survived 6 h in BALF. Strains 483 and 484 were able to grow faster compared to T17 and Strain 10 (Figure 4B).

A GRAM staining of the isolates after 2 h growth in BALF showed longer chains in T17, whereas other strains were similar (Figure 4C).

As differences in the growth of the isolates were detectable, subsequent functional tests for interaction of *S. suis* with components of the immune system were performed to find explanations for the sudden death of the pigs at the zoo.

Returning to the findings of NETs in the inner organs of the dead pigs, we tested the possibility that NET induction/a diminished DNase correlated with severity of disease, and whether diminished DNase activity in the isolates could explain the DIC. Therefore, in a subsequent step, we investigated the NET induction ability of the strains. Interestingly, only Strain 10 induced NETs to a significant degree. As *S. suis* DNases degrade NETs after 30 min of pre-incubation [37], NET fibers were no longer detectable after 3 h of incubation. Nevertheless, high numbers of neutrophils showed activation for NET release after 3 h of incubation in the case of Strain 10 (Figure 5A,B). All analyzed strains showed DNase activity (Figure 5C). Therefore, only a few NET fibers could be detected, as seen in the positive control with β-methly-cyclodextrin. Activation of neutrophils or NET formation was not detected after incubation with Strains 483 and 484. Thus, the DIC findings in the dead animals could not be explained by the NET induction ability of Strains 483 and 484. 

As the NET induction rate did not directly explain the DIC, we wondered whether the strains survived differently in the presence or absence of antibodies. Therefore, we compared the survival of the isolates in a reconstituted whole blood assay based on two different pig stocks (“TiHo” and “Zoo”). The Zoo stock, consisting of sera from pigs older than 2 months, reflected the survival of the different strains in the pig stock where the sudden death of several pigs had occurred. The TiHo stock reflected blood donors older than 2 months at the University of Veterinary Medicine Hannover (commonly abbreviated to TiHo) (Figure 6). 

These assays demonstrated that all tested strains, 483, 484, 10, and T17, either survived or were killed in the reconstituted blood of individuals from both stocks (Figure 6A). To identify whether the survival of Strains 483 and 484 in the reconstituted blood depended on host factors, we assigned the results from Figure 6A of the Zoo stock to two age groups. Indeed, significant differences were detectable for Strains 10 and T17 between the two age groups (Figure 6B). The reconstituted blood of some fattening pigs from the Zoo stock did not kill Strains 483 and 484 (ST28 *S. suis*). These findings might suggest that some pigs in the zoo had insufficient protective immunity, yet this remains to be analyzed. 

Finally, as *S. suis* is a zoonotic agent, we investigated whether the isolates from the zoo posed an increased risk for humans. However, as seen in Figure 6C, Strain 10 was able to survive in all blood donors, whereas the other isolates were mainly killed after 2 h of infection. Free DNA as a marker for NET release was analyzed in the plasma at the end of the whole blood assay. Interestingly, the donor with a high survival rate for Strain 10 showed the lowest amount of free DNA, independent of which *S. suis* isolate was used (Figure 6D). These findings reflected the individual characteristics of neutrophils from different donors as an additional influencing factor of the test outcome. In this case, it can be hypothesized that less NET release might favor the survival of Strain 10 in the blood.

## 3. Discussion

The starting point of the investigation described in this study was a disease outbreak in a small swine population at a zoo in the north-western part of Germany. In this swine herd, pigs were kept under extensive husbandry conditions, ensuring a high animal welfare level with a low pig density and the freedom for pigs to carry out species-specific behavior, including explorative behavior (Figure 1). These husbandry conditions are completely different from the intensive husbandry found on conventional farms. On conventional farms, key factors for infection dynamics with *S. suis* at herd level are known to be of high importance for disease pathogenesis due to the high pig density: (i) newborn piglets are colonized by vertical transmission during farrowing and horizontall transmission immediately after birth [50], (ii) until weaning, many piglets are carriers of *S. suis* on their tonsils, spreading the agent by comingling after weaning in the nursery unit [51], and (iii) any stressors during or after weaning trigger the development of disease [17]. It is hypothesized that the transition from colonization and persistence of *S. suis* in tonsillar tissue to the infection of preliminary sterile body compartments is only in part dependent on the virulence of strains, being mainly triggered by host factors (stress hormones) and co-infections [52]. In contrast, it was expected that in this clinical case under extensive husbandry conditions, trigger factors might be of lower importance for disease pathogenesis than bacterial virulence-associated factors. Bacterial isolation rates from tonsils of pigs of different age groups in this stock were lower than the published prevalence in healthy pigs (see Section 2.1) [22]. For cultural diagnostics, all different colony types suspicious for *S. suis* were chosen for further characterization. Only single suspicious colonies belonging to one or two colony types were identifiable. It can be expected that sensitive molecular diagnostics would have also led to the higher detection rate of the agent in this herd.

The *S. suis cps* type 2 strains responsible for disease in this clinical study could not be isolated from the tonsillar tissue of healthy animals (see Section 2.1.). The disease picture itself was different from most outbreaks on conventional farms. The presence of the pathogen in all inner parenchymatous organs was accompanied by disseminated intravascular coagulopathy (DIC) as a consequence of systemic disease (Figure 2). Findings were similar to those found in streptococcal toxic shock syndrome (STSS) in humans, which has not been described so far in pigs under non-experimental situations [53]. Zoonotic strains causing toxic shock syndrome in humans belong to serotype 2 and various clonal complexes (CCs). They were either *epf*-positive or -negative and the capsule was considered to be the major zoonotic determinant [53]. The *cps* type 2 strains isolated from diseased pigs in our study were *sly^−^,* and *epf*^−^, and therefore differed from pathogenic European serotype 2 *sly*^+^, *mrp*^+^, and *epf*^+^ strains mainly belonging to sequence type 1 [54] (Table 2 and Appendix A, Figure A1). Unexpectedly, the isolates in our study showed no characteristics in vitro that could be linked to high virulence with the exception of their ST28 (Figure 4, Figure 5 and Figure 6). In North America, ST28 strains are most prevalent with a percentage of more than 50% among all virulent serotype 2 strains [55,56]. A complex ST28 population structure has been reported, revealing at least five different clades and important virulence differences between genetic groups [57]. Recently, the ST28 was found to be the most frequent ST in the CC28, containing the most isolates with a pathogenic pathotype in a study in the United States [15]. In human patients suffering from infection with ST28, similar signs of disease as in the pigs in our case report with DIC and septicemia have been reported [58]. It has to be kept in mind that classification of strains by MLST is limited with respect to prediction of virulence and for phylogenetic analysis [57]. Based on the DIC findings and NETs in the histological samples, we performed assays like NET induction in this study, but were unable to produce further insight into the pathomechanisms behind the clinical picture of DIC (Figure 5). The different growth behavior of the isolates originating from lung and spleen belonging to the same Zoo strain in THB medium might indicate spontaneous mutations in the genomes of the strains, which had occurred during the course of infection or isolation procedures (Figure 4A). It cannot be ruled out that the lung strain was originally a contaminant strain, and might not have been the strain producing the clinical disease. NET induction was not observed in the strains examined in this study (Figure 5), so that this feature could not be directly correlated to development of DIC. However, NET induction is only one pathomechanism causing the cytokine storm responsible for the development of DIC [46]. Suilysin in *S. suis* serotype 2 strains involved in STSS was found to trigger excessive release of IL-1β followed by detrimental inflammation processes [59]. As the pathogenic Zoo strain described in this study was negative for suilysin, other bacterial factors in combination with a lack of an adequate host immune response might have led to DIC. The pro-inflammatory activity of other bacterial components was shown to be mediated by TLR2 recognition and ERK 1/2 MAPK phosphorylation, with the consequence of the excessive inflammation that is a hallmark of *S. suis* serotype 2-related disease [60]. Due to a lack of diagnostics in living animals suffering from septic shock, we were unable to confirm a cytokine storm in the dead pigs. Nevertheless, all negatives in the functional and molecular bacterial diagnostic findings in this case might suggest the lack of adequate adaptive humoral immunity in this stock as a key factor for disease exacerbation. While the commensal strain T17 was mainly killed by sera from all pigs (Figure 6A), this was not the case for the Zoo strains. This might be indicative of insufficient exposure and interaction of the immune system in susceptible pigs with the pathogenic strain before the time point of infection, or might be related to intrinsic bacterial factors. A clear age effect was seen with serum from the zoo pigs on the bacterial survival factor, revealing a high susceptibility in pigs of two months of age (Figure 6B). Adjusting the immunity of individuals in a stock to the same level can be achieved by vaccination, but production of autologous vaccine was not feasible for this stock due to the low number of animals. As an alternative, animals were exposed to saliva-contaminated ropes; this, however, presented a constant risk of disease in the younger exposed animals. The fact that *Haemophilus parasuis* also impacted herd health on this farm suggested external trigger factors for disease development. *Haemophilus-parasuis-*related disease outbreaks often emerge after transportation or mixing of pigs, both of which had happened in this herd. The fact that in some dead pigs no pathogen was cultivable could indicate that death was caused by this sensitive pathogen, which is scarcely cultivable if swine have already been dead for some hours. It can be hypothesized that other pre-conditions for disease outbreak existed in these extensive husbandry conditions. Due to low pig density and comparably low transition of *S. suis* between individuals after the decline in maternal antibodies, active immunization might be inadequate. The age group of diseased pigs in our study differed from that seen in conventional pig farms, with the highest disease incidence in weaners starting at 6 weeks of age and with a decrease in susceptibility at higher ages [19,50]. The much longer lactation period in our zoo herd and a hypothesized late contact with the agent might be the reason for older pigs becoming ill. Transporting the pig to the fattening site was a significant stressor, which coincided with exposure to a high bacterial load in the new environment due to a lack of adequate cleaning and disinfection measures. The positive effect of disinfectants for reducing *S. suis* infection is due to rapid killing of the agent by all conventional disinfectants [61]. 

The negative impact of low biosecurity has been well studied on commercial farms in different parts of the world [62]. In extensive pig husbandry, adequate biosecurity measures are difficult to apply. Good conditions for animal welfare in these systems are unable to compensate for a lack of biosecurity. In a recent study, the importance of *S. suis* infection in smallholder pig systems in Africa, combined with a high co-infection rate with PRRSV, *A. pleuropneumoniae*, Porcine Circovirus 2, *M. hyopneumoniae*, SIV, and porcine parvovirus was highlighted [20]. These findings support our hypothesis concerning disease pathogenesis in our case study. More than 73% of clinically healthy pigs were positive for *S. suis* on the African farms, and a multivariate logistic regression model resulted in a significant decrease in seroreagents associated with the use of disinfectants on farms, as well as in pigs older than six months. In our case, it was hypothesized that transport stress and unknown co-infecting agents are trigger factors that, acting coincidentally with the high susceptibility of pigs exposed to a high environmental load of the pathogen, might have ultimately led to exacerbation of the disease on this farm.

In conclusion, reconstituted whole blood assays from the serum samples from pigs of different age groups helped to reconstruct and understand the reasons for the sudden death of older pigs in the zoo stock. Reconstitution with serum from the zoo stock revealed good protection against tonsillar strains, but varying protection of individuals against pathogenic *cps* type 2 strains, especially in pigs of younger age. Additional characterization of causative *S. suis* strains in each case by sequence typing could therefore support selection of appropriate autologous vaccine candidates. An analysis of functional properties of strains would be even more useful to enhance the probability of protective effects of vaccines. A direct correlation between strain characteristics, NETs, and DIC was not observed. However, it is conceivable that the isolates from the zoo in some individuals led to septicemia, followed by a cytokine release, which could occur with NET induction and/or DIC. 

## 4. Materials and Methods 

### 4.1. Pathological and Histological Diagnostic of Dead Pigs

All deceased pigs were brought for routine diagnostics, including microbiological examination, to the Food and Veterinary Institute in Oldenburg or the Field Station for Epidemiology of the University of Veterinary Medicine Hannover, Foundation, Hannover. A routine gross pathological examination of the whole carcass was performed and all findings were recorded. Tissue samples for histological examination were fixed in 10% phosphate-buffered formaline for 48 h. After dehydrating samples in a graded series of ethanol, they were embedded in paraffin wax using xylene and cut into 8 µm sections with a motor-driven rotary microtome. Slides were deparaffinedzed and hydrated through descending concentrations of ethanol. Sections were stained with hematoxylin–eosin (HE, hemalaun after Delafield) following a routine histology protocol. Histology slices were examined with a standard light microscope (Olympus BX 41).

### 4.2. Collection of Diagnostic Samples from Pigs at the Zoo

At the zoo, tonsillar brushing and blood samples from the vena jugularis were taken from living pigs following routine veterinary sampling techniques. Pigs were immobilized with a conventional snare at the upper jaw. The mouth was kept open by a retractor so that the tonsilla palatina could be brushed with a disposable human interdental brush. The brush was immediately stored in a sterile plastic tube. Within 6 h of sampling, all samples were transported to the lab for cultural diagnostic. 

### 4.3. Microbiological Analysis of Samples from Pigs 

Organ tissue samples from necropsied pigs (Section 4.1) and tonsillar brushing specimens (Section 4.2) were examined by routine standard microbiological methods. For primary isolation, organ samples were inoculated in parallel on the following media: Columbia agar with 5% sheep blood (Merck GmbH, Darmstadt, Germany), Gassner agar (Oxoid Deutschland GmbH, Wesel, Germany), and staphylococci–streptococci-selective agar (Oxoid GmbH). For samples from the respiratory tract, chocolate blood agar (Columbia agar with 10% defibrinated horse blood, Oxoid GmbH) containing nicotinamide adenine dinucleotide (NAD, Merck GmbH), *Pasteurella*-selective agar (Oxoid GmbH), and Bordet Gengou agar (Becton Dickinson (BD) GmbH, Heidelberg, Germany) were also used. Tonsillar brushing specimens were cultivated only on CNA agar with 5% sheep blood (blood agar with colistin and nalidixic acid), Becton, Dickinson and Company (BD), Sparks, USA). All agar plates were incubated at 37 ± 1 °C for 48 h under standard atmospheric conditions with the exception of chocolate blood agar, which was incubated in a 5–8% CO_2_ atmosphere. For cultural diagnostics, all different colony types suspicious for *S. suis* were chosen for further characterization. While culture of organ samples from dead pigs resulted in high-grade pure culture of one colony type of *S. suis*, culture of tonsillar brushings resulted only in single suspicious colonies, belonging to one or two colony types. Suspicious single bacterial colonies were subcultivated after 24 and/or 48 h to generate pure cultures for further typing by their cultural and biochemical properties. *S. suis* was identified either by MALDI TOF mass spectrometry (MALDI Biotyper, Bruker Daltonic GmbH, Bremen, Germany) or by typical biochemical characteristics (hemolysis, Voges–Proskauer reaction, starch hydrolysis, growth on bile esculin agar at 37 and 42 °C) based on Reference [63], following routine diagnostic protocols. Pure cultures of *S. suis* were stored in Cryobanks (Cryobank, Mast Group Ltd., Bootle, United Kingdom) at −80 °C immediately after isolation.

### 4.4. Genotypic Analysis of Isolated Strains

Virulence-associated gene profiling was conducted as previously described [12]. Briefly, *S. suis* isolates were confirmed and characterized by multiplex PCR based on the detection of genes for capsular synthesis (*cps*); 1, 2, 4, 7, and 9 for suilysin (*sly*); for muramidase-released protein (*mrp*), including analysis for *mrp* variants; for extracellular factor (*epf*), including analysis for *epf* variants; for glutamate dehydrogenase (*gdh*); and for arginine deiminase (*arcA*), and visualized on agarose gel. 

For genotyping, bacteria were processed as previously described [64].

MLST was performed as previously described [14] with modifications regarding the primers for the genes *aroA* [64] and *mutS* [65].

The large *epf* variants were detected in a monoplex PCR assay as previously described [12,66].

All PCR reactions were conducted with OneTaq Quick-Load DNA Polymerase (New England Biolabs GmbH, Frankfurt am Main) in accordance with the manufacturer’s instructions (at a total volume of 50 µL).

To differentiate between serotype 2 and 1/2, Sanger sequencing was conducted as described previously [47]. The alignment with *cps2K* (GenBank: AF118389.1) was conducted with Clone Manager 9 Professional Edition.

### 4.5. Cultivation of Bacteria

*S. suis cps* type 2 strain 10 [28] and the three isolates from pigs from the zoo (Table 2) were smeared on Columbia agar plates with 7% sheep blood (Oxoid Deutschland GmbH) and incubated for 18–20 h at 37 °C. An overnight culture was prepared in 10 mL Bacto^TM^ Todd Hewitt Broth (THB) (Becton, Dickinson and Company) in a T406-2ACultubes™ (Simport^®^ Scientific Inc., Belœil, Canada) and incubated for 17–18 h. A 1:50 dilution was created in a 50 mL falcon tube with pre-warmed THB and the culture was incubated at 37 °C. The optical density (OD_600nm_) was measured every hour for 7 h with a spectrophotometer (Jenway 6310, Keison Products Ltd., Chelmsford, UK) to determine growth curves. 

For TECAN growth experiment, colony material was dissolved in 0.9% NaCl and adjusted to an optical density (OD_625nm_) between 0.05 and 0.11. This bacterial solution was diluted 1:100 in THB media and 100 µl was cultivated in a 96 u-bottom plate sealed with a foil. The plate was incubated in a TECAN plate reader for 20 h at 35°C. The optical density (OD_620nm_) was measured. 

### 4.6. Cryostock Preparation for In Vitro Experiments 

The bacteria were grown on Columbia blood agar and cultured afterwards in THB, as described in Section 4.5 until the beginning of the stationary growth phase (OD_600nm_ 1.2 ± 0.1). Cultures were mixed with sterile glycerol to a final concentration of 15% and 500 µL aliquots, immediately shock frosted in liquid nitrogen, and then stored at −80 °C. The colony-forming units per mL (CFU/mL) of the cyrostocks were determined after freezing by plating serial dilutions on blood agar plates. The CFU/mL of bacteria was checked every three months.

### 4.7. Ethical Statement for Collection of BALF, Porcine Blood, and Human Blood

The collection of fresh heparinized blood from healthy pigs was approved by the Lower Saxony State Office for Consumer Protection and Food Safety (LAVES), Germany under no. 12A243 and 18A302.

The collection of fresh heparinized blood from healthy human donors was approved by the Ethics Committee of the Hannover Medical School (MHH), Hannover, Germany and registered under no. 3295-2016.

Blood was collected in lithium–heparin tubes (S-Monovette^®^ 9ml LH, SARSTEDT AG & Co. KG, Nümbrecht, Germany).

Bronchoalveolar lavage fluid (BALF) was collected from fresh lungs extracted from healthy euthanized pigs. The euthanasia of these pigs was approved and registered by the local Animal Welfare Officer in accordance with the German Animal Welfare Law under number TiHo-T-2019-14. 

### 4.8. Collection of BALF and Growth of S. suis in BALF

One lung lobe was lavaged after separation from the carcass with 400 mL of sterile PBS to collect BALF as previously described [67]. The BALF was centrifuged (400× *g* at room temperature) and frozen in aliquots at −80 °C.

For the growth of *S. suis*, the BALF of one animal was used in three independent runs. For all four strains, fresh cryostocks were used (Section 4.6).

The total amount of 1.5 × 10^5^ CFU/mL was added to 200 µL of BALF and incubated for up to 6 h on a rotator at 7 rpm at 37 °C. At each time point, serial dilutions were plated on blood agar plates to determine the CFU/mL on the following day. All plates were incubated for 20 h at 37 °C. 

From each sample, 10 µL was dried on a glass slide and, after heat fixation, GRAM staining was conducted. Pictures were taken with an ApoTome microscope (Carl Zeiss AG, Oberkochen, Germany; Objective: Plan-Apochromat 63x/1.40 Oil DIC M27) and the Axiocam 105 color.

### 4.9. Staining of NETs in Histological Slices

Paraffin-embedded organs from Section 4.1 were cut into 4 µm sections and stretched on SuperFrost Plus™ adhesion slides (Thermo Fisher Scientific GmbH, Germany). Samples were deparaffinized and handled as previously described [68]. Briefly, the samples were incubated at room temperature for 1 h with primary antibodies diluted in blocking buffer. Histone–DNA complexes were stained with mouse anti DNA–histone-1 (Millipore MAB3864, stock 0.55 mg/mL, 1:100 diluted) and rabbit anti-*S. suis* antibody ([69] 1:500 diluted). Respective isotype controls were used to validate background fluorescence intensity. As secondary antibodies, a goat anti-mouse antibody (Dye Light488 conjugated highly cross-absorbed, Thermo Fisher Scientific GmbH; diluted 1:500 in blocking buffer) and goat-anti rabbit (Alexa633, Thermo Scientific GmbH; diluted 1:500 in blocking buffer) were used. After washing, each slide was covered with two drops Roti^®^-Mount FluorCare Prolong Gold (Carl Roth GmbH & Co. KG, Karlsruhe, Germany) with DAPI and a cover slip. Immunofluorescence samples were recorded using a Leica TCS SP5 AOBS confocal inverted-base fluorescence microscope with a HCX PL APO 40 × 0.75–1.25 oil immersion objective. Settings were adjusted with control preparations using an isotype control antibody. For the 3D picture, z-stacks were collected and analyzed with LAS X 3D Version 3.1.0 software from Leica. Z-stack pictures were used and the background was set to black using standard software settings. 

### 4.10. Purification of Porcine Neutrophils and NET Induction Assay

Porcine neutrophils were purified using Biocoll (1.077 g/mL; Merck Millipore Inc., Burlington, MA, USA) and hypotonic lysis of erythrocytes as previously described [70]. Cells were resuspended in Roswell Park Memorial Institute (RPMI) 1640 (no phenol red, Thermo Fisher Scientific (Bremen) GmbH).

Cover slips (8 mm; Thermo Fisher Scientific (Bremen) GmbH) were coated in accordance with the manufacturer’s instructions with poly-L-lysine (0.01% solution, Sigma Aldrich Inc., St. Louis MO, USA) for 20 min and washed three times with LPS-free 1× PBS. One hundred microliters of freshly isolated blood derived porcine neutrophils (2 × 10^5^ cells per well) was seeded in a 48 well plate with slides. As a negative control, RPMI only, and as a positive control, β-methyl-cyclodextrin (Sigma, final concentration 10 mM) were added to the neutrophils at a volume of 100 µL. *S. suis* cryostocks (Section 4.6) were used to infect neutrophils in separate wells with MOI 10 (2 × 10^6^ per well). Each sample was prepared in duplicate. The plate was centrifuged (370 × *g*, 5 min) and incubated at 37 °C, 5% CO_2_. After 3 h, samples were fixed with 4% paraformaldehyde (final concentration). 

NETs were stained as previously described [32]. Briefly, after blocking and permeabilization, neutrophils were incubated with a mouse monoclonal antibody (IgG2a) for DNA–histone 1 (MAB3864; Millipore 0.55 mg/mL, diluted 1:1000) and rabbit anti-human myeloperoxidase (Dako A0398, 3.2 mg, 1:300) for one hour at room temperature. As secondary antibodies, a goat anti-mouse antibody (Dye Light488 conjugated highly cross-absorbed, Thermo Fisher Scientific (Bremen), GmbH; diluted 1:500 in blocking buffer) and goat-anti rabbit antibody (Alexa633, Thermo Fisher Scientific (Bremen) GmbH; diluted 1:500 in blocking buffer) were used. Samples were recorded using a Leica TCS SP5 AOBS confocal inverted-base fluorescence microscope with a HCX PL APO 40 × 0.75–1.25 oil immersion objective.

For each sample, six randomly selected images per independent experiment were acquired and used for quantification of NET producing cells or activated cells (NET induction). The cells were counted using ImageJ software (version 1.52p, National Institute of Health, Bethesda, MD, USA).

### 4.11. DNase Activity Assay

Supernatants from *S. suis* cultures were collected after 6 h of growth curves (4.5.) by centrifugation (2100× *g*, 5 min at room temperature) and stored at −20 °C until usage. 

DNase activity of *S. suis* serotypes was tested in 50 µL supernatant mixed with 50 µL DNase buffer (pH 7.4, 3 mM CaCl_2_, 3 mM MgCl_2_, 300 mM TRIS) to a final reaction volume of 100 µL. In this sample, 0.5 µg calf thymus DNA (Sigma Aldrich) was incubated. The negative control was THB medium; as positive control, micrococcal nuclease from *Staphylococcus aureus* (Sigma Aldrich) was used.

Samples were incubated for 6 h at 37 °C. Visual examination of DNase activity was conducted in all assays after incubation with 1% agarose gel electrophoresis (100 V, 30 min) and staining of DNA with Roti- GelStain (Roth GmbH & Ko. KG, Karlsruhe, Germany). The gels were analyzed with a Bio-Rad MP Imaging System.

### 4.12. Reconstituted Whole Blood Assay with Porcine Blood

Heparin blood from healthy donor pigs from the TiHo stock was taken and blood cells were washed as follows: The blood was centrifuged (5 min, 1000 × *g*, room temperature, no brake) and the supernatant removed with a sucking pump. Afterwards, the remaining cells were washed twice with sterile 0.85% sodium chloride solution and centrifuged again (5 min, 1000 × *g*, room temperature, no brake). After sucking off the supernatant, the cells were re-suspended in 0.85% sodium chloride and the volume adjusted to the original blood volume. Next, 500 µL cell suspension was added to a 1.5 mL tube and centrifuged (5 min, 1000 × *g*, room temperature). Equal amounts of supernatant were taken off (250 µL) and the blood cells were dissolved with 250 µL cell-free blood components collected beforehand from the Zoo and TiHo stocks. Each tube with reconstituted blood was infected with respective *S. suis* strains from cryostocks (see Section 4.6) with approximately 1.5 × 10^5^ bacteria in 500 µL. Samples were incubated on a rotator (7 rpm, 37 °C, 2 h). At time points 0 and 2 h, the CFU/mL was determined by plating serial dilutions on blood agar plates. Based on the CFU/mL, the survival factor was calculated for 2 h (CFU_2h_ / CFU_0h_).

### 4.13. Whole Blood Assay in Human Blood

Blood was taken from three healthy donors. Per strain, 500 µL fresh heparinized blood was added to a 1.5 mL tube. The blood was infected with *S. suis* strains from cryostocks (Section 4.6.; infection dose per tube (mean): strain 10 = 1.6 × 10^5^ CFU; 483 = 1.75 × 10^5^ CFU; 484 = 2.0 × 10^5^ CFU; T17 = 6.0 × 10^4^ CFU). Tubes were incubated on a rotator (7 rpm, 37 °C, 2 h). The survival factor was calculated for 2 h as described in Section 4.12.

### 4.14. Pico Green Assay

To determine free DNA in plasma of samples from 4.13, a Pico Green assay (Invitrogen) was conducted in accordance with the manufacturer’s recommendation. A dilution series of calf thymus DNA (1, 0.1, 0.01, 0.001, 0 µg, Sigma Aldrich) was used as standard row. Samples were mixed 1:2 in a black, flat-bottom 96 well plate with Pico Green (prediluted 1:200 in TE buffer (1 0mM Tris- HCl, 1 mM EDTA). Afterwards, the plate was incubated for 5 min in the dark and then measured in a plate reader (TECAN Reader, filter 485/535, gain optimized, 25 flashes per well, multiple reads per well, shaken for 5 s). The values of DNA were calculated based on the standard row using GraphPad Prism 7.0 and 8.1 software.

### 4.15. Statistical Analysis

Data were analyzed using Excel 2016 (Microsoft) and GraphPad Prism 7.0 and 8.1 (GraphPad Software). Normal distribution of data was verified by Kolmogorov–Smirnov normality test or Shapiro–Wilk test for groups n = 3 (GraphPad software) prior to statistical analysis. Differences between groups were analyzed as described in the figure legends (* *p* < 0.05, ** *p* < 0.01, *** *p* < 0.001, **** *p* < 0.0001).

## Figures and Tables

**Figure 1 pathogens-08-00249-f001:**
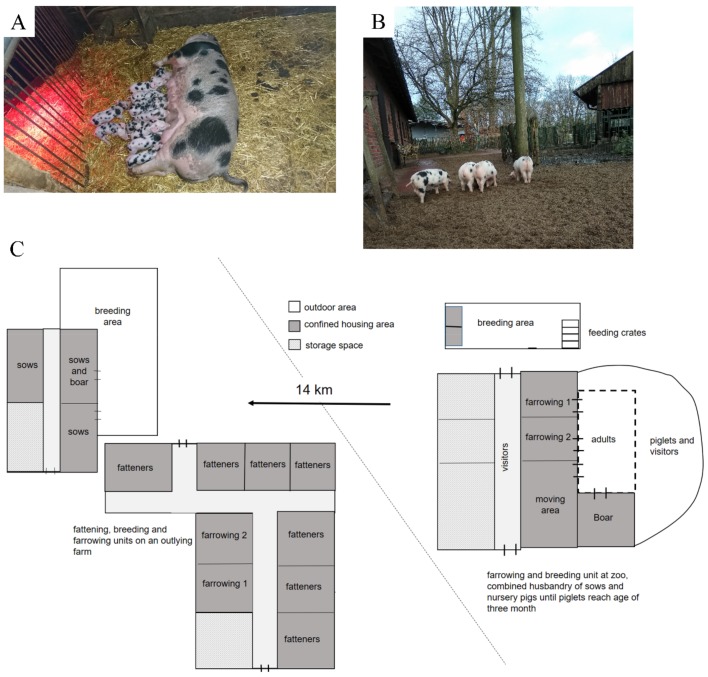
(**A**) The ancient and regional pig breed Buntes Bentheimer. (**B**) An impression of the extensive husbandry conditions, with low pig density and access to a sandy outdoor area at the zoo. Nursery pigs are shown. (**C**) The pig flow from zoo (right) with one boar and three sows with their litters to fattening unit (left). Here, an additional area for breeding (four sows and one boar) and farrowing exists.

**Figure 2 pathogens-08-00249-f002:**
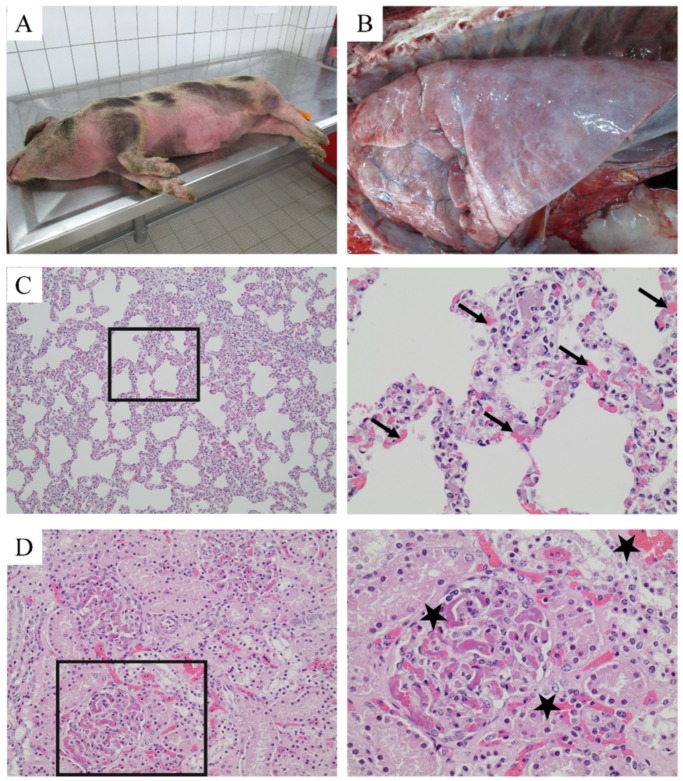
Overview of pathological and histological findings: (**A**) A generalized cutaneous hyperemia was present in the pigs that died suddenly. (**B**) Macroscopic changes in the lungs were found. The histological examination showed characteristics for a disseminated intravascular coagulopathy in the lung (**C**) and kidney (**D**). An overview is presented in the left panel, and a zoomed in image in the right panel. Multiple hyaline, fibrinous thrombi in glomerular (arrows) and pulmonary capillaries (asterisks) were detected.

**Figure 3 pathogens-08-00249-f003:**
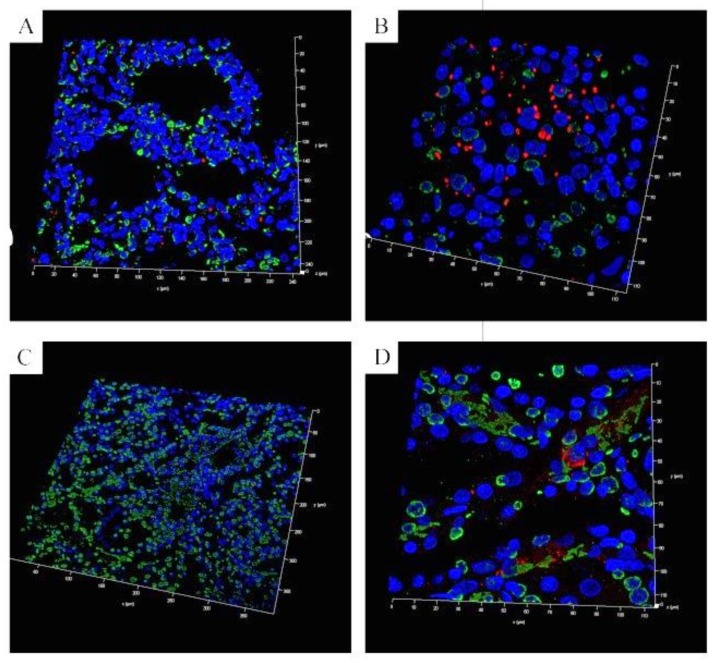
Immunofluorescence microscopic analysis of histological slices for neutrophil extracellular trap (NET) markers and *S. suis* (blue = DNA, green = DNA–histone-1 complexes = NETs, red = *S. suis*). The pictures show 3D images of the z-stacks. In the internal organs, characteristics of DIC and NET markers were detected in pigs which had died suddenly. (**A**) lung (z-stack present, 11.3 µm consisting of 91 sections), (**B**) spleen (z-stack present, 5.2 µm consisting of 32 sections), (**C**) kidney (z-stack present, 5.5 µm consisting of 34 sections) and (**D**) kidney (z-stack present, 8.3 µm consisting of 67 sections).

**Figure 4 pathogens-08-00249-f004:**
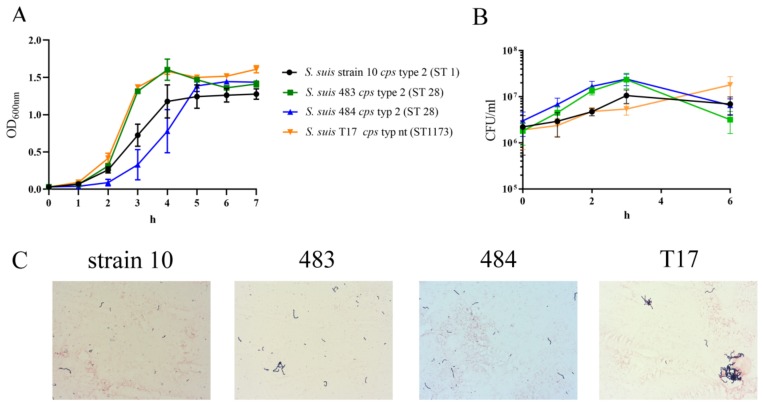
(**A**) The growth was analyzed by measuring the optical density over 7 h. (**B**) The CFU/mL of isolates growing in bronchoalveolar lavage fluid (BALF) of healthy pigs was monitored for 6 h. Data in A and B show the mean (±SEM) of at least three independent experiments (in B, time-points 3 and 6 h, n = 4; all other time-points, n = 7). Statistical differences in (A) were analyzed via two-way ANOVA (p_time_ < 0.0001, p_strain_ < 0.02) followed by Tukey’s multiple comparison test. A significant difference was detected after 3–4 h between all strains except Strains 483 and T17. Statistical differences in (B) were analyzed via mixed model two-way ANOVA (p_time_ < 0.0009, p_strain_ = 0.1) followed by Tukey’s multiple comparison test. A significant difference was detected after 3 h between Strain 483 and T17, Strain 483 and Strain 10, Strain 484 and T17, and Strain 484 and Strain 10. (**C**) GRAM staining showed chain formation of the isolates after 2 h growth in BALF.

**Figure 5 pathogens-08-00249-f005:**
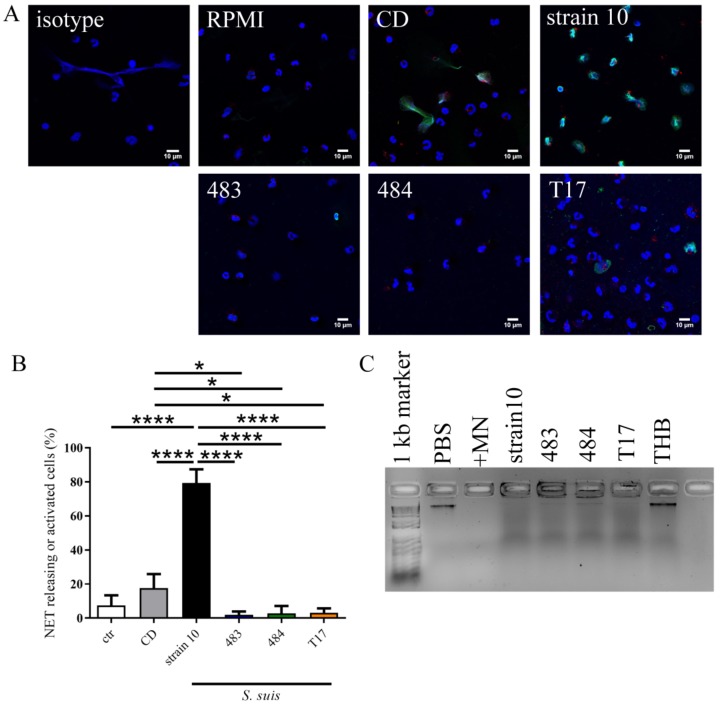
(**A, B**) NET-releasing neutrophils and neutrophils activated for NET release were analyzed 3 h after *S. suis* infection of primary porcine neutrophils. The analysis was conducted via immunofluorescence microscopy. Cells with a clear off-shoot of DNA (green/blue) were counted as NET-releasing cells. Cells with a signal for DNA–histone-1 staining (green) and changes in nuclei (size and shape) were counted as activated cells; cells without changes in nuclei were counted as negative (blue = DNA (Hoechst), green = DNA–histone-1 complexes = NETs, red = myeloperoxidase). Representative pictures are shown from three independent experiments (the mean of six pictures per sample and experiment was used for statistical calculations). Roswell Park Memorial Institute 1640 medium (RPMI) was used as negative control (ctr). β-methyl-cyclodextrin (CD) was used as positive control. Settings were adjusted with control preparations using a respective isotype control antibody instead of the specific primary antibody. Statistical analysis of NET-releasing cells was done via one-way ANOVA followed by Tukey’s multiple comparison test (*p* values: * *p* < 0.05, ** *p* < 0.01, ** *p* < 0.001, *p* < 0.0001). The bars represent mean ± SD. (**C**) DNase activity assay showed activity in all tested strains. Micrococcal nuclease (+MN) was used as positive control. Phosphate buffered saline (PBS) and Todd Hewitt Broth (THB) were used as negative controls.

**Figure 6 pathogens-08-00249-f006:**
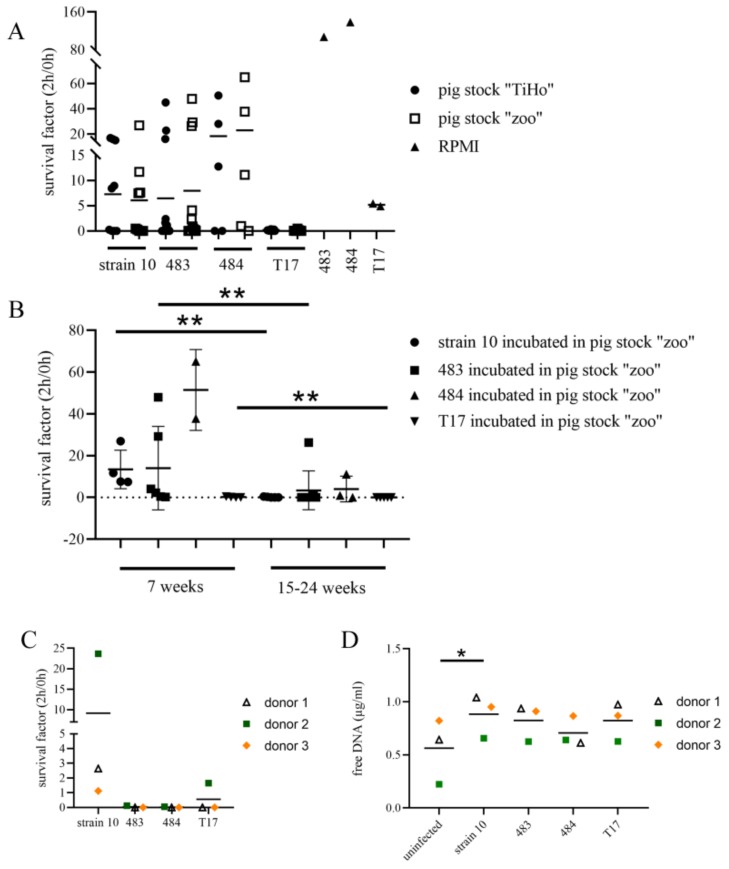
(**A**) Reconstituted whole blood assay in porcine blood: Each dot represents the serum of one individual that was used in each graph for the tested strains in the same experiment. The mean is presented in all graphs. In (A), blood cells originating from five different donors were used. RPMI (Roswell Park Memorial Institute 1640 medium) with blood cells was used as a growth control without serum. No significant difference was detected by one-way ANOVA (no matching). (**B**) Data from (A) of *S. suis* strains incubated in the reconstituted blood from the Zoo stock, the grouping thereof conducted by age. Statistical analysis was done with Mann–Whitney test between each strain and both age groups. (**C**) The survival of the different strains in whole blood of humans was analyzed. Each dot represents a human individual. No significant difference was detected with one-way ANOVA. (**D**) Supernatants from the whole blood assay in (C) were analyzed for free DNA by Pico Green. A non-infected blood sample was used from each donor to determine non-specific released DNA during incubation. The detected free DNA at time-point 0 was subtracted from the presented values. Significant difference was detected with one-way ANOVA (* *p* = 0.0042) followed by Dunnett’s test.

**Table 1 pathogens-08-00249-t001:** Diagnostic findings in dead pigs (3–7-month-old pigs) including identification of *Streptococcus suis.*

Date	Clinical findings	Pathological findings	Bacteriological findings	Diagnosis
10/18	sudden death, 38.5 kg	disseminated intravascular coagulopathy	*S. suis* (lung, heart, spleen, kidney)	systemic disease
10/18	sudden death, 33.5 kg	disseminated intravascular coagulopathy	*S. suis* (lung, heart, spleen, kidney)	systemic disease
12/18	sudden death, 47 kg	fibrinopurulent carpitis, fibrinopurulent leptomeningitis, hemorrhagic gastritis and jejunitis	no isolate	treated (antibiotics)
12/18	sudden death, 103 kg	fibrinopurulent epi- and pericarditis, leptomeningitis	no isolate	treated (antibiotics)
12/18	sudden death, 38 kg	purulent leptomeningitis, pleuritis, epicarditis, hemorrhagic gastritis, jejunitis	*Haemophilus parasuis* ST 13 (brain, lung, heart)	Glaesser’s disease

**Table 2 pathogens-08-00249-t002:** Genotypic characterization of isolated strains (*epf** included testing for variants). The isolate T28 was not further analyzed for its sequence type (ST) (X).

Name	Date	Origin	Capsular type	*mrp*	*sly*	*epf**	*gdh*	*arcA*	ST
483	10/18	pig 1, lung	2	+	-	-	+	+	28
484	10/18	pig 2, spleen	2	+	-	-	+	+	28
T17	01/19	pig 3, tonsil	n.t.	-	-	-	+	+	1173
T28	01/19	pig 4, tonsil	n.t.	-	-	-	+	+	x

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
