# Peer review of "From Stable to Lab—Investigating Key Factors for Sudden Deaths Caused by Streptococcus suis"

_pathogens, 2019, doi:10.3390/pathogens8040249_

Round 1
Reviewer 1 Report
Major comments
Because S. suis infections in fattening pigs are also an increasing problem in my country, the case reported in this manuscript is very interesting for me. The fact that the causative strains were ST28 is also interesting; therefore, it is worth publishing this case. However, because only a few isolates were investigated in lab, and no characteristics that can explain the sudden death of the fattening pigs were found, I feel that this manuscript is too long for its contents. In addition, I have many questions about this study as described below.
Lines 35-36 (Protection against … vaccine candidate selection) and lines 370-372 (Protection against … selection of vaccine candidates): I cannot understand logics of this discussion. Reword these sentences to be easily understood.
Lines 135-137: The authors found three positive samples from healthy pigs, but only one tonsil isolate (T17) was investigated (Table 2). Why didn’t the authors investigate the other two isolates?
Lines 135-137: I think many readers would like to know whether sows of the dead pigs carried ST28 S. suis strains or not. Did the authors attempt to isolate S. suis from sows?
Lines 138-143: Did the oral fluid sample contain S. suis? After immunization, did anti-S. suis serotype 2 antibody titers increase in the immunized sows and younger piglets?
Lines 181-193 and Figure 4AB:
- Were differences in the growth of S. suis strains statistically significant?
- Please add error bars in Figures 4A and 4B.
- Why did the authors investigate S. suis growth in BALF only for 2 hours? If the authors continue the tests for 7 hours (like Figure 4A) or more, what happens?
Lines 245-246 (Therefore, … in the zoo.): The meaning of this sentence is hard to make out. I suggest following modification if the authors agree.
“These findings suggest that some pigs in the zoo did not have sufficient protective immunity against ST28 S. suis strains even after getting older, and this may have caused the death of fattening pigs in the zoo.”
Data in Figure 6 was statistically analyzed by the Student’s t-Tests. As the authors know, t-Test assumes that the data follows the normal distribution. Although the authors verified the data by Kolmogorov-Smirnov normality test (lines 552-553), I wonder if the data in Figure 6 really follows the normal distribution. So, I would like the authors to show the Kolmogorov-Smirnov normality test results.
Lines 315-317 (The different growth … the original environment.): Because the authors repeatedly subcultured the isolates using culture media for primary isolation, generation of pure culture and the growth tests, it is difficult to think that the different growth behavior of the isolates was due to an epigenetic metabolic imprint by the original environment. Although we cannot disregard an epigenetic metabolic imprint, spontaneous mutations occurred in the genome of the strains during the course of infection or during the course of isolation and purification may have caused the different growth behavior.
Minor comments
Line 43 (with 35 different serotypes): Strains of serotypes 20, 22, 26 and 32-34 have been reclassified into different Streptococcus species (DOI: 10.1016/j.vetmic.2005.01.003, doi: 10.1099/ijs.0.067116-0, DOI 10.1099/ijsem.0.002204), and a novel serotype Chz was proposed in 2015 (doi: 10.1128/AEM.02962-14); therefore, 35 serotypes do not reflect current taxonomical situation of S. suis. Line 129 (In a period of five months): Was the five months from Oct to Feb (i.e., autumn and winter seasons)? Line 165 and Table 2: “serotype 2” should be represented as “cps type 2” because the authors investigated the isolates only by PCR (i.e., investigated only genotype). Serotype is a phenotype. Tests using antiserum are necessary to determine serotypes. Figure 4C: Because colonies in the photos were too small, differences in colony morphology could not be confirmed by the photos. Provide high-power colony images or delete the photos. Please explain what “isotype”, “RPMI” and “CD” mean in the legend. Line 244: … for the survival OF THE STRAINS 483 AND 484 between … Figure 6C: Indicate strain name used for the experiments. Line 433: the culture WAS incubated… Line 514 (mixed 1:2 with): 1:1?? (because 50ul supernatant and 50 ul DNase buffer were mixed.)
Author Response
Answer to the reviewers pathogens 618955
Dear Editors and Reviewers,
Thank you for giving us the opportunity to revise our manuscript “From stable to lab – investigating key factors for sudden deaths caused by Streptococcus suis “
Please find below our answers to the comments and questions of the reviewers.
We have prepared a revised version of the manuscript, highlighting the changes using the Track Changes function in Microsoft Word, and we have prepared a point to point response in order to address the remarks out by the reviewer.
We followed the reviewer comments and modified figure 1, 4, 5 and 6, table 2 and included appendix D. Therefore we uploaded all figures with this revised manuscript version. Furthermore we upload a certificate for “Proofreading of attached scientific manuscript” by the English Editorial Office of our university.
We thank again the reviewers for the comments to our study and tried to improve the manuscript based on the constructive comments.
With kind regards
Nicole de Buhr
Reviewer 1
Because S. suis infections in fattening pigs are also an increasing problem in my country, the case reported in this manuscript is very interesting for me. The fact that the causative strains were ST28 is also interesting; therefore, it is worth publishing this case. However, because only a few isolates were investigated in lab, and no characteristics that can explain the sudden death of the fattening pigs were found, I feel that this manuscript is too long for its contents. In addition, I have many questions about this study as described below.
Answer: The manuscript was shortened as much as possible. Unfortunately additional examinations had to be performed within the review period, which must also be described and we were asked to rewrite the figure legends. We excluded some figures and respective results and followed the suggestions of the reviewers.
Lines 35-36 (Protection against … vaccine candidate selection) and lines 370-372 (Protection against … selection of vaccine candidates): I cannot understand logics of this discussion. Reword these sentences to be easily understood.
Answer: Both sentences have been reworded for a better understanding.
Lines 135-137: The authors found three positive samples from healthy pigs, but only one tonsil isolate (T17) was investigated (Table 2). Why didn’t the authors investigate the other two isolates?
Answer: T17 has been chosen primarily just as an example strain for a colonizing strain. The other two strains were less characterized than T17 so far. The other two isolates have been further characterized during this revision process and results are included now. Within the 10 days of the review period some functional assays have been performed also with these strains (please see revised table 2 and new appendix D).
Lines 135-137: I think many readers would like to know whether sows of the dead pigs carried ST28 S. suis strains or not. Did the authors attempt to isolate S. suis from sows?
Answer: In addition to the herd diagnostic in the fattening pigs we tried to isolate S. suis from the tonsils of all sows for further characterization, and we were successful in 2 sows to find alpha-haem. streptococci with typical biochemical properties for S. suis (2 suspicious isolates). We focused in further steps only onto the tonsillar isolates of the fatteners as described in the paper. Other experts told us that the approach in general to lock for ST28 might be senseless (to lock for “a needle in the hay”) by picking just visible colonies from the plate. They suggested us to use microbiome analysis/NGS – but as far as we know, we cannot differentiate sequence types by these methods. Because the study was not financially funded/supported, we restricted our diagnostics to the isolates from the fattening pigs. As soon as we were sure, that no S. suis cps type 2 was found on the tonsils of the fatteners, we did not perform further typing with the sows isolates. In all cultural samples from living pigs (tonsils and saliva) -although using selective media- high amount of staphylococci and enterococci and other bacterial species are grown. Colonies suspicious for S. suis were only found in samples from some individuals, but only 1-2 single colonies on one plate visible.
Lines 138-143: Did the oral fluid sample contain S. suis? After immunization, did anti-S. suis serotype 2 antibody titers increase in the immunized sows and younger piglets?
Answer: Cultural diagnostic in oral fluids from the fattening pigs was not successful (it was not possible to culture S. suis, because enterococci, staphylococci and other alpha-häm. bacteria dominated in culture). We are quite sure, that S. suis is everywhere in the oral cavity of pigs and in saliva, but so far we are not able to culture them as frequently as we want to.
The routine diagnostic labs did not offer antibody tests for S. suis, because this test is not helpful for field diagnostics. During the review process, we asked several routine labs to do the diagnostic for us, but they are not able to. For this reason, it was not possible to perform the serological testing. Researchers, who developed an ELISA to detect specific antibodies against S. suis capsular type 2 came to the conclusion, that the CPS-ELISA cannot be used as a diagnostic tool to identify infected animals on farms, because they found no differences between sera of animals from herds with S. suis capsular type 2 associated disease and those from healthy herds (Reference: Martin de1 Campo Sepulveda, E., Altman, E., Kobisch, M., D’Allaire, S., Gottschalk, M. (1996): Detection of antibodies against Streptococcus suis capsular type 2 using a purified capsular polysaccharide antigen-based indirect ELISA. Veterinary Microbiology 52, 113-125).
Lines 181-193 and Figure 4AB:- Were differences in the growth of S. suis strains statistically significant?
Answer:
We conducted a 2 way ANOVA test and included the results in the manuscript. Furthermore a multiple comparisons analysis (Tukey) was done, but these results are not included in the figure as this in our opinion would make the graphic too crowded. A short summary is included in the results part and the figure legend.
Please add error bars in Figures 4A and 4B.
Answer: Error bars are added in Figure 4 A and 4B. In some points the Graph Pad software drwas no error bar as they are shorter than the symbol.
Why did the authors investigate suis growth in BALF only for 2 hours? If the authors continue the tests for 7 hours (like Figure 4A) or more, what happens?
Answer: The growth test in BALF was conducted for a longer time period. The material and methods section was adapted and results of the extended growth tests were included in the results section now, including a statistical analysis (Figure 4B).
Lines 245-246 (Therefore, … in the zoo.): The meaning of this sentence is hard to make out. I suggest following modification if the authors agree.
“These findings suggest that some pigs in the zoo did not have sufficient protective immunity against ST28 S. suis strains even after getting older, and this may have caused the death of fattening pigs in the zoo.”
Answer: The original sentence was deleted and the sentence was rewritten following the reviewers suggestion.
Data in Figure 6 was statistically analyzed by the Student’s t-Tests. As the authors know, t-Test assumes that the data follows the normal distribution. Although the authors verified the data by Kolmogorov-Smirnov normality test (lines 552-553), I wonder if the data in Figure 6 really follows the normal distribution. So, I would like the authors to show the Kolmogorov-Smirnov normality test results.
Answer: We thank the reviewer for this comment and tested again all data in Figure 6 for normal distribution. Based on the results we changed in some of the graphs the statistics as described in the new figure legend. The results of the Kolmogorov-Smirnov normality test and other normality test’s available in Graph Pad software are added with this submission (see end of PDF file).
The statistical test’s were chosen based on experimental settings, statistical questions and normal distribution.
Lines 315-317 (The different growth … the original environment.): Because the authors repeatedly subcultured the isolates using culture media for primary isolation, generation of pure culture and the growth tests, it is difficult to think that the different growth behavior of the isolates was due to an epigenetic metabolic imprint by the original environment. Although we cannot disregard an epigenetic metabolic imprint, spontaneous mutations occurred in the genome of the strains during the course of infection or during the course of isolation and purification may have caused the different growth behavior.
Answer: Subculturing for repeated tests was always performed from a conserved frozen culture, which has experienced only 3 passages from the original sample. For other bacterial species the epigenetic metabolic imprint has already been shown by this method of culturing.
Nevertheless we followed the reviewer’s suggestion and deleted the hypothesis from the paper and directed the discussion more in spontaneous mutations of strains.
Line 43 (with 35 different serotypes): Strains of serotypes 20, 22, 26 and 32-34 have been reclassified into different Streptococcus species (DOI: 10.1016/j.vetmic.2005.01.003, doi: 10.1099/ijs.0.067116-0, DOI 10.1099/ijsem.0.002204), and a novel serotype Chz was proposed in 2015 (doi: 10.1128/AEM.02962-14); therefore, 35 serotypes do not reflect current taxonomical situation of S. suis.
Answer: The new references are implemented now and the paragraph was corrected (line 44-46).
Hill, J.E.; Gottschalk, M.; Brousseau, R.; Harel, J.; Hemmingsen, S.M.; Goh, S.H. Biochemical analysis, cpn60 and 16S rDNA sequence data indicate that Streptococcus suis serotypes 32 and 34, isolated from pigs, are Streptococcus orisratti. Vet. Microbiol. 2005,107,63-9.
Nomoto, R.. Maruyama, F.; Ishida, S.; Tohya, M.; Sekizaki, T.; Osawa, R. Reappraisal of the taxonomy of Streptococcus suis serotypes 20, 22 and 26: Streptococcus parasuis sp. nov. Int. J. Syst. Evol. Microbiol. 2015, 65, 438-43.
Tohya, M1.; Arai, S.; Tomida, J.; Watanabe, T.; Kawamura, Y.; Katsumi, M.; Ushimizu, M.; Ishida-Kuroki, K.; Yoshizumi, M.; Uzawa,Y.; Iguchi, S.; Yoshida, A.; Kikuchi, K.; Sekizaki, T. Defining the taxonomic status of Streptococcus suis serotype 33: the proposal for Streptococcus ruminantium sp. nov. Int. J. Syst. Evol. Microbiol. 2017, 67, 3660-5.
Pan, Z.; Ma, J.; Dong, W.; Song, W.; Wang, K.; Lu, C.; Yao, H. Novel variant serotype of streptococcus suis isolated from piglets with meningitis. Appl. Environ. Microbiol. 2015, 81, 976-85.
Line 129 (In a period of five months): Was the five months from Oct to Feb (i.e., autumn and winter seasons)?
Answer: The reviewer is right. Death cases occurred from October to February. This was added to the text now.
Line 165 and Table 2: “serotype 2” should be represented as “cps type 2” because the authors investigated the isolates only by PCR (i.e., investigated only genotype). Serotype is a phenotype. Tests using antiserum are necessary to determine serotypes.
Answer: The expression “serotype 2” is deleted from the manuscript and replaced by “cps type 2”.
Figure 4C: Because colonies in the photos were too small, differences in colony morphology could not be confirmed by the photos. Provide high-power colony images or delete the photos.
Answer: Because the manuscript should either be shortened, we deleted the photos, methods description and results part about the colonies.
Please explain what “isotype”, “RPMI” and “CD” mean in the legend.
Explanations are given in the legend of figure 5 now (lines 260-272).
Line 244: … for the survival OF THE STRAINS 483 AND 484 between …
Answer: The sentence was corrected following the suggestions of the reviewer.
Figure 6C: Indicate strain name used for the experiments.
Answer: The strain name was added in the figure.
Line 433: the culture WAS incubated…
Answer: The grammar was corrected following the suggestion of the reviewer
Line 514 (mixed 1:2 with): 1:1?? (because 50ul supernatant and 50 ul DNase buffer were mixed.)
Answer: The description was corrected.

Reviewer 2 Report
This manuscript seeks to identify phenotypical characteristics and/or molecular explanations attributable to Streptococcus suis isolates from infections resulting in recent sudden deaths of pigs. Two isolates taken from organs of different dead pigs, strains 483 and 484, were compared to a tonsillar strain (T17) taken from a healthy pig and to a reference meningitis strain (strain 10). Genotype and phenotype analyses were conducted and included documenting the presence of a subset of virulence genes, strain growth rates, colony and cell morphologies, the ability to induce NETosis, and strain survival in blood. Although no definitive factors could be attributed to strains 483 and 484 that would explain the severe pathologies seen in the pig outbreak, the manuscript will provide important information documenting these events and the new S. suis isolates. Although I find this report to be an important contribution to the S. suis literature, the manuscript’s current state of English grammar is unacceptable for publication, particularly in the Results section and for figure legends. I strongly recommend that editing be conducted by someone who’s native language is English, as there are many places throughout the manuscript in which the meaning of sentences are unclear. I point a few out below, but many more sentences are in need of grammatical revision. However, overall, as far as I can understand what was conducted, the work appears to be of sound scientific methodology.
Areas of extra importance:
Abstract, 3rd sentence is unnecessary (“For the host interaction…”) Line 56, unsure what physiological pig’s mucosal colonizer means. Line 112. ‘all-in-all-out’ not understood Line 136 (Tonsillar brushings…). This is where strain T17 should be introduced. Line 183, ‘genotypically identical picture of the isolates’ is unclear Line 207, “Data in A and B are showing the mean of three independent experiments.” should be included in figure 4 legend. Line 209, “we proofed the hypothesis”. As it is not possible to prove a hypothesis (can only disprove a hypothesis), you should say you tested the possibility whether NET induction correlated with severity in disease. Line 217, rephrase to say, “Activation of neutrophils as indicated by NET formation, was not detected when incubated with strains 483 and 484…” Figure 5A. The top 7 panels (lower magnification) are very difficult to see, even when zoomed in on the pdf file. Can these be made brighter? Alternatively, are these top 7 panels even needed? It seems the lower 7 panels are much easier to see the results. Figure 5 legend, describe what CD is (B-methyl-cyclodextrin) and that it is the positive control. Could the lack of NET formation actually account for the enhanced virulence seen in strains 483 and 484? Line 237, “was not going in line with” does not make sense. Lines 243-246 (final 2 sentences of 2nd to last paragraph of results). These need to be clarified—they currently do not make sense. Final sentence of Results, “Interestingly,…” This sentence is unlcear…please elaborate what is meant. MOST IMPORTANT: Figure 6. This figure and its legend are completely unclear. Graph labels need to be included for panels C (x-axis?), D (why are there 2 types of symbols for each strain?), E and F (why are there 3 symbols for each strain?). The figure legend needs careful editing, as it now does not make much sense.
Author Response
Answer to the reviewers pathogens 618955
Dear Editors and Reviewers,
Thank you for giving us the opportunity to revise our manuscript “From stable to lab – investigating key factors for sudden deaths caused by Streptococcus suis “
Please find below our answers to the comments and questions of the reviewers.
We have prepared a revised version of the manuscript, highlighting the changes using the Track Changes function in Microsoft Word, and we have prepared a point to point response in order to address the remarks out by the reviewer.
We followed the reviewer comments and modified figure 1, 4, 5 and 6, table 2 and included appendix D. Therefore we uploaded all figures with this revised manuscript version. Furthermore we upload a certificate for “Proofreading of attached scientific manuscript” by the English Editorial Office of our university.
We thank again the reviewers for the comments to our study and tried to improve the manuscript based on the constructive comments.
With kind regards
Nicole de Buhr
Reviewer 2
1 Although I find this report to be an important contribution to the S. suis literature, the manuscript’s current state of English grammar is unacceptable for publication, particularly in the Results section and for figure legends. I strongly recommend that editing be conducted by someone who’s native language is English, as there are many places throughout the manuscript in which the meaning of sentences are unclear. I point a few out below, but many more sentences are in need of grammatical revision. However, overall, as far as I can understand what was conducted, the work appears to be of sound scientific methodology.
Answer: The paper was revised by an English native speaker. A certification about proofreading of the manuscript by an English native speaker is attached.
Abstract, 3rd sentence is unnecessary (“For the host interaction…”)
Answer: The sentence was deleted from the abstract.
Line 56, unsure what physiological pig’s mucosal colonizer means.
Answer: An explanation about physiological colonizers of mucosa in pigs is given (lines 60-63). More details can be seen in the reference Kernaghan et al. 2012.
Line 112. ‘all-in-all-out’ not understood
Answer: An explanation about the meaning of “all-in-all-out” in pig production is given (lines 123-125).
Line 136 (Tonsillar brushings…). This is where strain T17 should be introduced.
Answer: As suggested “tonsillar sampling” was replaced by “tonsillar brushing”, and T17 and the two other tonsillar strains, which had been typed now, are introduced now in this paragraph (line 154-155).
Line 183, ‘genotypically identical picture of the isolates’ is unclear.
Answer: The sentence was formulated in a new way.
Line 207, “Data in A and B are showing the mean of three independent experiments.” should be included in figure 4 legend.
Answer: The explanation according to data A and B was already implemented in line 212, but due to a wrong formatting, not direct in the figure legend visible. Therefore we reorganized the figure legend.
Line 209, “we proofed the hypothesis”. As it is not possible to prove a hypothesis (can only disprove a hypothesis), you should say you tested the possibility whether NET induction correlated with severity in disease.
Answer: The sentence was formulated in a new way following the suggestion of the reviewer.
Line 217, rephrase to say, “Activation of neutrophils as indicated by NET formation, was not detected when incubated with strains 483 and 484…”
Answer: The sentence was formulated in a new way following the suggestion of the reviewer.
Figure 5A. The top 7 panels (lower magnification) are very difficult to see, even when zoomed in on the pdf file. Can these be made brighter? Alternatively, are these top 7 panels even needed? It seems the lower 7 panels are much easier to see the results.
Answer: Because the manuscript should either be shortened as requested by reviewer 1, we deleted the top 7 panels. We try always to give an overview to show not only single events. In the uploaded tiff files the quality of the top panels was good.
Figure 5 legend, describe what CD is (B-methyl-cyclodextrin) and that it is the positive control.
Answer: The legend to figure 5 was complemented by meanings of abbreviations.
Could the lack of NET formation actually account for the enhanced virulence seen in strains 483 and 484?
Answer: This can be a hypothesis, but we have not further investigated this yet. Lack of NET formation could be explained by virulence factors as described for other bacteria:
Cell Host Microbe. 2008 Aug 14;4(2):170-8. doi: 10.1016/j.chom.2008.07.002.
The IL-8 protease SpyCEP/ScpC of group A Streptococcus promotes resistance to neutrophil killing.
Zinkernagel AS1, Timmer AM, Pence MA, Locke JB, Buchanan JT, Turner CE, Mishalian I, Sriskandan S, Hanski E, Nizet V.
Front Cell Infect Microbiol. 2018 Jul 9;8:235. doi: 10.3389/fcimb.2018.00235. eCollection 2018.
The Staphylococcus aureus Extracellular Adherence Protein Eap Is a DNA Binding Protein Capable of Blocking Neutrophil Extracellular Trap Formation.
Due to the request of reviewer 1 to shorten the manuscript, we did not include this interesting hypothesis/aspect into the discussion.
Line 237, “was not going in line with” does not make sense.
Answer: The sentence was formulated in a new way.
Lines 243-246 (final 2 sentences of 2nd to last paragraph of results). These need to be clarified—they currently do not make sense.
Answer: The respective sentences were formulated in a new way.
Final sentence of Results, “Interestingly,…” This sentence is unlcear…please elaborate what is meant.
Answer: An additional explanation for this observation was given.
MOST IMPORTANT: Figure 6. This figure and its legend are completely unclear. Graph labels need to be included for panels C (x-axis?), D (why are there 2 types of symbols for each strain?), E and F (why are there 3 symbols for each strain?). The figure legend needs careful editing, as it now does not make much sense.
Answer: Figure 6 was edited in a new way and the legend was elaborated. We hope that meaning of symbols is now clearer.

Round 2
Reviewer 1 Report
Major comments
The revised manuscript is easier to understand than original one. To make the manuscript more reader friendly, I suggest some additional modifications.
Lines 34-35 (These findings … selection.): I suggest following modifications.
These findings highlight the benefit of further characterization of the causative strains in each case by sequence typing before autologous vaccine candidate selection.
Table 2: Because isolate T31 is negative for gdh, this isolate may not be S. suis.
Figure 5A: Please indicate which color (blue, green, light blue, red) represents which phenomenon (NET fiber, activated neutrophils, etc) in the legend of Figure 5.
Lines 234-236 (Thus, … in the dead animals): I suggest following modifications.
Thus, the DIC findings in the dead animals could not be explained by the NET induction ability of the strains 483 and 484.
Lines 263-265 (These findings suggest … at the zoo.): The findings that support this suggestion are not explained in the text. Before this sentence, explain the results of ST28 S. suis strains shown in Figure 6A (i.e., describe that the reconstituted blood of some fattening pigs did not kill ST28 S. suis strains (483 and 484) efficiently).
Lines 388-389 (Additional … candidates): I suggest following modifications.
Additional characterization of causative S. suis strains in each case by sequence typing can therefore support selection of appropriate autologous vaccine candidates.
Appendixes A and C: Add information of positive control samples in the legends.
Minor comments
Italicize “cps” in “cps type”. Line 121: disseminated intravascular coagulopathy -> DIC Line 179: virulence factor genes -> virulence-associated genes Lines 199-200 (significant differences in growth were detectable at 3 and 4h): Which pairs did the author detected significant differences in? Between 484 and strain 10? Line 207: Delete “In”. Line 285: … detected with … -> … detected between “Zoo” and “TiHo” stocks with … Lines 320-321: clonal complexes -> clonal complexes (CCs) Line 453: for six hours -> for seven hours ??? Lines 460-461: The authors measured OD600nm of bacteria grown on Columbia blood agar. Is it possible? Lines 537-538 (THB … in parallel.): Delete this sentence, because the same information appears in lines 541-542. Lines 561 and 567: Delete (SF).
Author Response
Dear Editors and Reviewers,
We thank the reviewer’s for the second report and the constructive suggestions.
Please find below our answers to the comments and questions of the reviewers.
We have prepared a revised version of the manuscript, highlighting the changes from first revision and second revision using the Track Changes function in Microsoft Word, and we have prepared a point to point response in order to address the remarks out by the reviewer.
We thank again the reviewers for the comments to our study and tried to improve the manuscript based on the constructive comments.
With kind regards
Nicole de Buhr
Reviewer 1
Major comments
The revised manuscript is easier to understand than original one. To make the manuscript more reader friendly, I suggest some additional modifications.
Lines 34-35 (These findings … selection.): I suggest following modifications.
These findings highlight the benefit of further characterization of the causative strains in each case by sequence typing before autologous vaccine candidate selection.
Answer: We followed the reviewer modification suggestion.
Table 2: Because isolate T31 is negative for gdh, this isolate may not be S. suis.
The reviewer is right, we discussed this as well before the resubmission. As this is a typical phenomenon in daily diagnostic, we have not deleted the information’s but included now a comment to the results for the reader (line 197 ff).
Figure 5A: Please indicate which color (blue, green, light blue, red) represents which phenomenon (NET fiber, activated neutrophils, etc) in the legend of Figure 5.
Answer: We included information’s about the counting and the colour in the figure legend. As the criteria for NET releasing and NET activated cells are not only due to a different colour, we included the criteria as well (e.g. nuclei form).
Lines 234-236 (Thus, … in the dead animals): I suggest following modifications.
Thus, the DIC findings in the dead animals could not be explained by the NET induction ability of the strains 483 and 484.
Answer: We thank the reviewer for this very good suggestion and changed the sentence.
Lines 263-265 (These findings suggest … at the zoo.): The findings that support this suggestion are not explained in the text. Before this sentence, explain the results of ST28 S. suis strains shown in Figure 6A (i.e., describe that the reconstituted blood of some fattening pigs did not kill ST28 S. suis strains (483 and 484) efficiently).
Answer: We thank the reviewer for this suggestion and included a sentence before.
Lines 388-389 (Additional … candidates): I suggest following modifications.
Additional characterization of causative S. suis strains in each case by sequence typing can therefore support selection of appropriate autologous vaccine candidates.
Answer: We followed the suggestion and changed the sentence.
Appendixes A and C: Add information of positive control samples in the legends.
Answer: We included informations’s in the legend’s. All reference strains were obtained from other laboratories as specified in the Acknowledgements of the revised manuscript and confirmed through independent typing by our own laboratories.
Minor comments
Italicize “cps” in “cps type”.
Answer: We changed it in the whole manuscript.
Line 121: disseminated intravascular coagulopathy -> DIC
Answer: We corrected the manuscript.
Line 179: virulence factor genes -> virulence-associated genes
Answer: We corrected the manuscript.
Lines 199-200 (significant differences in growth were detectable at 3 and 4h): Which pairs did the author detected significant differences in? Between 484 and strain 10?
Answer: We included information in the main text as described similar in the legend.
Line 207: Delete “In”.
Answer: We corrected the manuscript.
Line 285: … detected with … -> … detected between “Zoo” and “TiHo” stocks with …
Answer: We corrected the manuscript.
Lines 320-321: clonal complexes -> clonal complexes (CCs)
Answer: We corrected the manuscript.
Line 453: for six hours -> for seven hours ???
Answer: We corrected the manuscript.
Lines 460-461: The authors measured OD600nm of bacteria grown on Columbia blood agar. Is it possible?
Answer: The reviewer is right, information’s were missing. We have measured in THB, and therefore we corrected the manuscript.
Lines 537-538 (THB … in parallel.): Delete this sentence, because the same information appears in lines 541-542.
Answer: We corrected the manuscript.
Lines 561 and 567: Delete (SF).
Answer: We corrected the manuscript.
Reviewer 2 Report
I commend the authors for the substantial improvement.
Author Response
Dear Editors and Reviewers,
We thank the reviewer’s for the second report and the constructive suggestions.
Please find below our answers to the comments and questions of the reviewers.
We have prepared a revised version of the manuscript, highlighting the changes from first revision and second revision using the Track Changes function in Microsoft Word, and we have prepared a point to point response in order to address the remarks out by the reviewer.
We thank again the reviewers for the comments to our study and tried to improve the manuscript based on the constructive comments.
With kind regards
Nicole de Buhr
Reviewer 2
I commend the authors for the substantial improvement
Answer: we thank the reviewer for the reports and the improvement of the manuscript.